# Functional Express Proteomics for Search and Identification of Differentially Regulated Proteins Involved in the Reaction of Wheat (*Triticum aestivum* L.) to Nanopriming by Gold Nanoparticles

**DOI:** 10.3390/ijms26157608

**Published:** 2025-08-06

**Authors:** Natalia Naraikina, Tomiris Kussainova, Andrey Shelepchikov, Alexey Tretyakov, Alexander Deryabin, Kseniya Zhukova, Valery Popov, Irina Tarasova, Lev Dykman, Yuliya Venzhik

**Affiliations:** 1K.A. Timiryazev Institute of Plant Physiology, Russian Academy of Sciences, Moscow 127276, Russia; anderyabin@ifr.moscow (A.D.); zhukova@ifr.moscow (K.Z.); popov@ifr.moscow (V.P.); 2V.L. Talrose Institute for Energy Problems of Chemical Physics, N.N. Semenov Federal Research Center of Chemical Physics, Russian Academy of Sciences, Moscow 119334, Russia; kusainova7531@gmail.com (T.K.); iatarasova@yandex.ru (I.T.); 3The Russian State Center for Animal Feed and Drug Standardization and Quality, Moscow 123022, Russia; admin@sevin.ru (A.S.); tretyakov81@gmail.com (A.T.); 4Institute of Biochemistry and Physiology of Plants and Microorganisms, Saratov Scientific Centre of the Russian Academy of Sciences, Saratov 410049, Russia

**Keywords:** *Triticum aestivum*, nanopriming, gold nanoparticles, ultrafast proteomics, mass spectrometry, photosynthetic apparatus, pro-/antioxidant balance

## Abstract

Proteomic profiling using ultrafast chromatography–mass spectrometry provides valuable insights into plant responses to abiotic factors by linking molecular changes with physiological outcomes. Nanopriming, a novel approach involving the treatment of seeds with nanoparticles, has demonstrated potential for enhancing plant metabolism and productivity. However, the molecular mechanisms underlying nanoparticle-induced effects remain poorly understood. In this study, we investigated the impact of gold nanoparticle (Au-NP) seed priming on the proteome of wheat (*Triticum aestivum* L.) seedlings. Differentially regulated proteins (DRPs) were identified, revealing a pronounced reorganization of the photosynthetic apparatus (PSA). Both the light-dependent reactions and the Calvin cycle were affected, with significant upregulation of chloroplast-associated protein complexes, including PsbC (CP43), chlorophyll a/b-binding proteins, Photosystem I subunits (PsaA and PsaB), and the γ-subunit of ATP synthase. The large subunit of ribulose-1,5-bisphosphate carboxylase/oxygenase (RuBisCo) exhibited over a threefold increase in expression in Au-NP-treated seedlings. The proteomic changes in the large subunit RuBisCo L were corroborated by transcriptomic data. Importantly, the proteomic changes were supported by physiological and biochemical analyses, ultrastructural modifications in chloroplasts, and increased photosynthetic activity. Our findings suggest that Au-NP nanopriming triggers coordinated molecular responses, enhancing the functional activity of the PSA. Identified DRPs may serve as potential biomarkers for further elucidation of nanopriming mechanisms and for the development of precision strategies to improve crop productivity.

## 1. Introduction

Abiotic stressors that negatively impact plant growth and crop productivity include heavy metals, drought, salinity, and cold and hot temperatures. Effective techniques paired with technical developments are required for crop production systems to be robust in the face of abiotic environmental constraints. The application of nanotechnology in agriculture has gained attention in recent years, especially for the eradication of various abiotic stressors [1,2,3]. Nanomaterials had shown promising results for alleviating the harmful impacts of abiotic stressors by modulating plant physiological parameters at multiple levels of plant organization. Nanomaterials are already widely used as pesticides, fertilizers, and for water and soil purification [4,5,6]. It should be noted that intensive use of NPs raises concerns about their possible accumulation in ecosystems. Concerns are heightened in agriculture, where soils are deliberately exposed to products containing NPs, such as substances with pesticide or antibacterial activity, as well as irrigation with untreated wastewater [7]. Soil, which is one of the final sinks of NPs, can represent a source of NPs entering food webs. Therefore, the issue of developing strategies for the use of NPs in experimental biology and agriculture is highly relevant. In this direction, a number of important aspects should be taken into account: (1) the stability and chemical nature of NPs, (2) dose-effect, and (3) the need to test on different objects [8]. Nanopriming—the process of seed treatment in solutions containing nanoparticles (NPs) before planting—is one of the innovative technologies used to manage plant metabolism and increase stress tolerance [8,9,10]. This technology is considered economically viable and environmentally friendly for biological systems [10,11]. It is important that, during nanopriming, the metal particles present in the colloidal solution do not enter the above-ground parts of the plants, or do so only in trace amounts. Plants grown from primed seeds differ from the control group in terms of enhanced growth rates, photosynthetic activity, and higher resistance due to the regulation of the pro-/antioxidant balance [8,9,10]. Among metal NPs, gold nanoparticles (Au-NPs) occupy a special place because they have unique physicochemical properties. Au-NPs are capable of plasmon resonance—high electron activity on the particle surface at a certain wavelength [12,13]. Thanks to this property, Au-NPs modulate photosynthetic processes and the antioxidant system (AOS) of plants. On the other hand, due to their chemical inertness, Au-NPs are considered safe for living cells and are widely used in biomedical research [14]. The molecular mechanisms underlying the action of NPs on plant organisms remain poorly understood. One approach to studying them may be the analysis of cell and tissue proteomes using accurate ultrafast mass spectrometry-based quantitative analysis aimed at identifying regulatory proteins involved in the response to nanopriming. Functional express proteomics is a powerful tool for identifying DRPs and assessing their role in physiological and biochemical processes [15]. Unlike genomics and transcriptomics, proteomic analysis allows characterizing the dynamics of the protein profile in response to external stimuli, including nanopriming. This approach allows us to understand the molecular basis of the action of NPs and opens up new opportunities for the development of promising biotechnological approaches to increasing the adaptive potential of plants.

In this study, for the first time, a comprehensive proteomic analysis of wheat seedlings was performed after nanopriming of seeds with a colloidal solution of Au-NPs in order to identify regulatory proteins involved in the response of plants to the action of NPs. The results of the proteomic analysis were confirmed by data on ultrastructural changes in mesophyll cell organelles, as well as by physiological and biochemical methods for determining the activity of the photosynthetic apparatus (PSA) and the pro-/antioxidant balance.

## 2. Results

The diameter of synthesized Au-NPs was determined by absorption spectroscopy, transmission electron microscopy (TEM), and dynamic light scattering methods (Figure 1). Measurements of hydrodynamic radius and zeta potential of the particles were performed on a Zetasizer Nano-ZS (DLS, Malvern Instruments, Malvern, UK). Electron-microscopic studies were carried out on a Libra 120 digital electron microscope (Carl Zeiss, Oberkochen, Germany). UV-vis double-beam spectrophotometer Specord 250 (Analytik Jena, Germany) was used to measure absorption and elastic scattering spectra. The maximum of the absorption spectrum of the obtained sol was λmax = 518 nm, and the optical density was A520 = 1.18. As shown in Figure 1a, ehe average diameter of the obtained nanoparticles was 15.4 nm. The number of particles in 1 mL at A520 = 1 was 1.6 × 1012. The zeta potential was −19 mV. The maximum concentration of gold in the solution was 57 µg/mL. The stock solution of nanoparticles was diluted with distilled water immediately before concentration tests and experiments. In the present study, we used a concentration of 20 μg/L of Au-NPs based on previous studies and the results of concentration tests.

Comparative proteomic analysis of shoots relative to roots in both the Au-NPs treatment group and the control group enabled the identification of DRPs, which were subsequently analyzed through gene ontology (GO) enrichment analysis. Figure 2a presents the biological processes identified in control plants and plants treated with Au-NPs. Notably, the processes related to photosynthesis (GO:0015979) and photosynthesis, light reaction (GO: 0019684) exhibited markedly higher GO enrichment scores in the Au-NPs treated group compared to the control group, with increases in approximately 2.3-fold and 3.3-fold, respectively. Conversely, the process of generation of precursor metabolites and energy (GO:0006091) demonstrated greater enrichment in the control group by approximately 1.3-fold relative to the Au-NPs treatment.

Moreover, plants treated with Au-NPs showed enhanced activity in biological processes associated with the regulation of intracellular pH homeostasis. These included vacuolar acidification (GO:0007035), intracellular pH reduction (GO:0051452), general pH reduction (GO:0045851), regulation of intracellular pH (GO:0051453), regulation of pH (GO:0006885) and regulation of cellular pH (GO:0030641). This suggests that nanopriming with Au-NPs may influence cellular ion balance and compartment acidification mechanisms, potentially contributing to improved stress tolerance and metabolic regulation.

The number of DRPs shown in Figure 2b for wheat seedlings in the control group and under the influence of Au-NPs differed only slightly: 144 upregulated proteins were identified in Au-NP-treated plants, while 147 upregulated proteins were found in the control group. It is worth noting that, in addition to the DRPs common to both groups, unique proteins were present exclusively in each group. Specifically, 90 unique DRPs were detected in the control group, whereas 87 unique DRPs were identified in the Au-NP-treated group.

Next, we analyzed individual proteins with the most pronounced differences in isoform number and expression levels related to the activity of the PSA: the biological processes Photosynthesis (GO:0015979), Photosynthesis, light reaction (GO:0019684), and Response to light intensity (GO:0009642). For example, in Au-NP-treated plants, the expression of three DRPs belonging to photosystem I (PS I) was higher than in the control group. These proteins—A0A3B6CD48, A0A3B5ZT69, and S4Z1Y0 (psaB photosystem I P700 chlorophyll an apoprotein A2)—are presented in Figure 3a. Nanopriming of seeds with Au-NPs also promoted increased expression of the PsaA protein (photosystem I P700 chlorophyll an apoprotein A1), also part of PS I, which was not expressed in control seedlings (S4Z9B3). Two chlorophyll proteins of 43 kDa (CP43 proteins: B2CHJ8, S4Z090), components of the PS II reaction center, were identified as expressed only in Au-NP-treated plants. Moreover, a large number of PS II stability factors were detected in Au-NP-treated plants, including high chlorophyll fluorescence 136 (HCF136: A0A3B6RQB5, A0A3B6U0E4), fibrillins (FBNs: A0A7H4LNP3, A0A3B6DLF1, A0A3B6B5T5, A0A3B6CE00), and MAR-binding filament-like protein 1 (MFP1: A0A3B6EG07), as shown in Figure 3b. Except for HCF136 (A0A3B6RQB5), these proteins were absent in the control group. In addition, nanopriming with Au-NPs influenced proteins involved in the chloroplast protein import machinery (Tic/Toc complex). Three translocons at the inner envelope membrane of chloroplasts 110 (TIC110 proteins: A0A3B5YX09, A0A3B5ZUP0, A0A3B5XYP5), shown in Figure 3c, were found to be unique to the Au-NP-treated plants. Furthermore, in contrast to the control group, Au-NP-treated plants showed increased expression of plastid transcriptionally active 16 proteins (PTAC16: A0A3B6ELF1, A0A077RU36, A0A3B6GV04), which are essential for plastid gene expression. In the control group, however, the presence of RuBisCO assembly factors (RAF1: A0A3B5ZTJ0 and A0A3B5YW66), involved in RuBisCO biogenesis, was detected. Additionally, enhanced activity of sedoheptulose-1,7-bisphosphatase—an enzyme involved in the regeneration of five-carbon sugars in the Calvin cycle and regulation of carbon flux—was observed, as shown in Figure 3c.

Additionally, Au-NP seed treatment led to a noticeable increase in the expression of several chlorophyll a-b binding proteins (CBPs), which are part of the chlorophyll-binding protein complexes of photosystem II in wheat seedlings, as shown in Figure 4a. However, the amount of CBPs was twice as high in control plants. Our study showed that nanopriming led to a significant increase in expression of non-specific serine/threonine protein kinase (PK: A0A3B6N3R0) and 50S ribosomal protein L12 (L12: A0A3B6GUD0). In addition, expression of elongation factor P (EF-P, A0A1D5XV56) is presented in Figure 4b. Control plants contained another elongation factor (EF-Tu, A0A3B6NQK6), but its expression was lower. EF-P and EF-Tu differ in their functions during translation in plants. The control plants showed a fairly high expression of two carbonic anhydrases (CA: A0A3B6FNB9, A0A3B6GWA0), which were absent in Au-NP-treated plants.

Changes in the proteome of wheat seedlings induced by Au-NPs nanopriming affected DRPs not only in the light-dependent but also in the Calvin cycle stage of photosynthesis, as shown in Figure 5. For instance, Au-NP treatment enhanced the expression of proteins P11383, B2CHJ6, and A0A3B6FN83, which belong to the large subunit of RuBisCO, as presented in Figure 5b. According to real-time PCR data, the expression of the RbcL gene also increased to a greater extent in Au-NP-treated plants compared to the RbcS gene (Figure 5c,d).

It is also noteworthy that Figure 6a presents RuBisCO activases whose expression was elevated in control plants (C6YBD7, A0A3B6JIJ3, A0A3B6JFS0, and A0A3B6IN76). Both treatment groups shared two common activase isoforms, while also exhibiting unique isoforms. The activases uniquely expressed in Au-NP-treated plants included A0A3B6CEF2, A0A3B6B6N9, and A0A3B6JFS0.

Our study showed that nanopriming led to increased expression of certain chloroplast ATP synthases in wheat seedlings, as shown in Figure 6b. In the control plants, the expression of five ATP synthase subunit alpha proteins—responsible for regulatory functions—was threefold higher compared to Au-NP-treated plants. Conversely, the expression of three ATP synthase subunit beta proteins, which contain the catalytic sites, was slightly elevated in Au-NP-treated wheat plants. Notably, the expression of two ATP synthase subunit gamma proteins was more than 15-fold higher in Au-NP-treated plants. Expression of mitochondrial ATP synthase subunit alpha (A0A3B6SN12) was also higher in control plants (Figure 6b). In Au-NP-treated plants, eight phosphoglycerate kinase (PGK) proteins were identified, whereas in control plants only three PGK proteins were detected, and their expression was four times lower. You can find these and other data not presented in the Appendix A.

Importantly, the proteomic changes observed in response to Au-NPs nanopriming were supported by physiological and biochemical analyses. Nanopriming did not affect the germination of seeds, but the leaf length of the treated plants was 13% longer than in the control variant (Table 1). As shown in Figure 7 and Table 1, nanopriming significantly altered the ultrastructure of mesophyll cells in wheat seedlings. Specifically, the area of peroxisomes and mitochondria decreased by 28% and 43%, respectively, under the influence of nanopriming, while the area of chloroplasts increased by 11%. Moreover, treatment with Au-NPs led to a 75% increase in starch grain area and a 9% increase in the number of grana within chloroplasts. Nanopriming also affected the distribution of grana according to the number of thylakoids: the proportion of grana containing 2–6 thylakoids increased by 16%, while the proportion of larger grana with 7–16 thylakoids decreased by 19% (Table 2).

As shown in Figure 8, nanopriming with Au-NPs did not affect the respiration rate, but increased the photosynthesis rate in wheat seedlings by 30%. At the same time, the chlorophyll content in wheat leaves was increased by 8%, and the carotenoid content was not changed under the influence of nanopriming. Au-NPs reduced the total content of soluble sugars (sucrose, fructose and glucose) in wheat leaves by 16%. It is important to note that no differences were found between the control and primed plants in the rate of lipid peroxidation, determined by the MDA content (Figure 9c).

Analysis of the data on the effect of nanopriming on the activity of antioxidant enzymes in wheat leaves showed that the control plants had a higher activity of the key antioxidant enzyme superoxide dismutase (SOD) than the treated plants. At the same time, nanopriming increased the activity of ascorbate peroxidase (APX) by 2 times, while there were no changes in the activities of CAT and POD (Figure 10).

Analysis of the gold content in various organs (seeds, leaves, roots) of wheat seedlings treated with Au-NPs showed that the seeds contained the highest amount of gold (3.8 mg/kg DW), while the metal content in the roots was 2 times lower. Trace gold content was noted in the leaves of plants grown from treated seeds (Table 3).

## 3. Discussion

The combined use of -omics approaches, such as transcriptomics and proteomics, has proven valuable in recent years for elucidating molecular mechanisms, including those associated with abiotic stress responses. Integrative transcriptomic and proteomic analyses have enabled the identification of mechanisms involved in the in planta reduction of Au or Ag NPs [16]. In another study [15], a characteristic molecular signature involving proteins related to photosynthesis and glycolysis regulation was proposed as a potential marker of the biotic effects of seed treatment with iron-based compounds, and this was further confirmed in subsequent studies. In research where metal nanoparticles exhibited toxic effects on plants, proteomic responses revealed enhanced expression of protective and stress-related proteins [17]. We have previously demonstrated a stimulatory effect of gold nanoparticle treatment on wheat growth, several biochemical parameters, and cold tolerance [18].

In the present study, we employed ultrafast liquid chromatography/MS profiling [19,20,21] for quantitative proteomic analysis of shoots and roots of 10-day-old spring wheat seedlings grown from Au-NP–primed seeds. Our results showed that in the shoots of Au-NP–treated plants, the majority of DRPs were associated with the following biological processes: Generation of precursor metabolites and energy (GO:0006091), Photosynthesis (GO:0015979), Photosynthesis, light reaction (GO:0019684), Response to light intensity (GO:0009642), Electron transport chain (GO:0022900), Proton transmembrane transport (GO:1902600), Anaerobic respiration (GO:0009061), Response to cadmium ion (GO:0046686), Response to metal ion (GO:0010038), and Energy derivation by oxidation of organic compounds (GO:0015980). These findings are, to some extent, consistent with previously published data.

### 3.1. Photosynthesis

The primary differences identified in the DRPs of wheat plants treated with gold nanoparticles (Au-NPs), compared to control plants, were predominantly associated with the processes of Photosynthesis (GO:0015979), Response to Light Intensity (GO:0009642), and specifically the Photosynthesis light reaction (GO:0019684). Notably, the expression levels of chlorophyll a-b binding proteins (CBPs), which play a critical role in light energy capture and its transfer to photochemical reaction centers, were elevated in Au-NP-treated plants (Figure 4a). The regulation of CBP expression is controlled by the circadian clock and is frequently employed as a marker of circadian rhythms in plants. Furthermore, Au-NP plants exhibited increased chlorophyll content (Figure 8c). CBPs are understood to facilitate the distribution of excitation energy between PS I and PS II [22]. Consequently, the augmented abundance of CBPs likely enhanced light energy absorption through the formation of a greater number of light-harvesting complexes (LHC)—multi-molecular assemblies of protein and chlorophyll molecules embedded in the thylakoid membranes of chloroplasts. Previous studies have demonstrated that the absence of chlorophyll b (Chl b) in ch1 Arabidopsis thaliana mutants results in reduced growth rates, diminished leaf size, and decreased biomass. These phenotypic effects were initially attributed solely to the impaired photosynthetic capacity due to Chl b deficiency. However, in addition to suppressing photosynthetic function, Chl b deficiency was found to delay flowering and prematurely initiate ontogenetic and induced senescence programs [23].

It is important to highlight that Au-NPs possess unique physicochemical properties, notably their capacity for plasmon resonance. This phenomenon enhances the activity of electrons on the nanoparticle surface through collective oscillations induced by light of specific wavelengths. The consequences of this include: (1) intensified energy transfer within LHC due to the ability of surface electrons to capture incident photons; (2) an increased population of excited electrons resulting in enhanced light absorption; and (3) rapid and efficient charge separation at the reaction centers, which accelerates electron transport, photophosphorylation, and oxygen evolution [12,13]. These properties of Au-NPs underlie their modulatory effects on photosynthetic processes.

Our results also demonstrated that the expression of the RuBisCo large subunit protein was threefold higher in Au-NP-treated plants compared to controls (Figure 5b). It is well established that in higher plants, the gene encoding the large subunit of RuBisCo, rbcL, is located in the plastid genome, whereas the gene for the small subunit, RbcS, is encoded by a multigene family in the nucleus. Although the active sites of RuBisCo reside on the large subunits, the expression of the small subunit regulates the overall RuBisCo pool size and can influence the catalytic efficiency of the enzyme complex [24]. Therefore, it can be hypothesized that Au-NP treatment predominantly affected plastid genome expression.

RuBisCo forms dead-end inhibited complexes by binding various sugar phosphates, including its substrate ribulose-1,5-bisphosphate. Rescue from this inhibited state is mediated by molecular chaperones belonging to the AAA + ATPase family, known as RuBisCo activases [25]. Interestingly, control plants exhibited a higher abundance of RuBisCo activase proteins, with expression levels approximately twice as high as those observed in Au-NP treated plants (Figure 6b).

Concurrently, two peroxisomal (S)-2-hydroxy-acid oxidases, or glycolate oxidase (GLO) proteins, were detected in Au-NP treated plants, whereas control plants possessed four such proteins (one GLO5 and three GLO1 isoforms). Overexpression of GLO1 increased GLO activity by 110% in rice leaves. Overexpression of GLO5 had no effect on GLO activity [26]. GLO5 functions as a photorespiratory enzyme that can significantly regulate photosynthesis, possibly by inhibiting Rubisco activase. GLO isoenzymes display diverse enzymatic properties and may fulfill distinct physiological roles in plants [26]. Glycolate oxidase is an important peroxisomal oxidase involved in photorespiration. Plant photorespiration begins with the oxygenating reaction of ribulose 1,5-bisphosphate carboxylase-oxygenase (RuBisCo) in chloroplasts. This process produces a toxic intermediate metabolite, phosphoglycolate (2-PG), which is further converted to glycolate by 2-PG phosphatase (PGP). Glycolate is transferred to peroxisomes and oxidized into glyoxylate by GLO with an equimolar amount of hydrogen peroxide (H_2_O_2_) released. In addition to its metabolic function in photorespiration, GLO has been reported to play roles in plant photosynthetic regulation and stress resistance. Suppression of GLO leads to glyoxylate accumulation and inhibits photosynthesis, while overexpressing GLO confers improved photosynthesis under high light and high temperature in rice. In the rice lines with GLO activities suppressed, the dwarfism phenotype was reduced and H_2_O_2_ content was also detected in these rice lines [27,28]. Thus, in Au-NP plants with only GLO5 with the function of inhibiting Rubisco activase, the role of the RuBisCo activation mechanism by RuBisCo activase was probably reduced. Nevertheless, Au-NP treatment was a more effective way to increase RuBisCo L expression compared to the mechanisms found in control plants (e.g., high expression of RuBisCo activase).

Supporting studies report upregulation of proteins related to protein synthesis and photosynthesis in nanoparticle-treated plants. For example, exposure to 40 mg L^−1^ TiO_2_ nanoparticles under biotic stress led to increased expression of defense- and photosynthesis-related proteins, with concurrent downregulation of those linked to protein metabolism and stress [29]. This enhancement in photosynthesis was attributed to elevated activity of key enzymes, including Rubisco. Similarly, Yang et al. [30] observed increased RuBisCo activity in spinach following TiO_2_ treatment.

Furthermore, another study [31] reported an increase in the Rubisco L, indicating that, at the observed time point, Calvin cycle activity was not diminished relative to controls. This may be attributed either to Rubisco’s oxygenation activity feeding into photorespiration or to an overcompensatory response of the cultivar to drought stress, as suggested by similar observed changes.

It is worth noting that enhancing the carboxylation efficiency of RuBisCO remains a promising strategy for improving photosynthesis and increasing crop yield. Typically, this enhancement is achieved through genetic transformation aimed at increasing the expression of the small subunit or by boosting the activity of RuBisCO activase. Based on our proteomic analysis, it can be assumed that a similar effect may be observed in plants grown from seeds primed with Au-NPs.

According to our data, the abundance of PS I proteins PsaA and PsaB, as well as the expression of two PS II proteins—CP43 reaction center protein, which is an integral antenna of PSII and may play a crucial role in repair after photodamage—were higher in Au-NP treated plants (Figure 3). The function of CP43 involves binding Chl a and β-carotene and transferring excitation energy to the reaction center, thereby increasing the light-harvesting capacity for photosynthesis. The collective data suggest an increase in the number of PSII complexes in Au-NP plants. This is further supported by ultrastructural data showing an increase in the amount of thylakoid membranes in chloroplasts (Figure 7, Table 1 and Table 2).

Moreover, the increase in chloroplast reaction center components (CP43, CBPs, PsaA, PsaB, ATP synthase subunit gamma) led to enhanced stability factors. For example, the expression of two PS II stability/assembly factors, HCF136, was twice as high in Au-NP plants compared to controls Figure 3b. HCF136 is known to be involved in the assembly of the PS II reaction center in plants. Mutants deficient in HCF136 fail to accumulate PSII complexes and suffer from photooxidative stress even under very low light intensities [32].

Our data also revealed that Au-NP-treated plants contained FBNs and MFP1. FBNs are highly conserved plastid lipid-associated proteins that play important roles in plant physiology. Four FBN proteins were detected in Au-NP-treated plants but were absent in control plants (Figure 3b). Notable functions of FBNs include maintaining plastid stability, promoting plant growth and development, enhancing stress responses—particularly drought tolerance—regulating carotenoid accumulation, protecting against oxidative stress, participating in chromoplast structure formation, and involvement in the storage, transport, and synthesis of lipid molecules, which provide photoprotective functions under high light stress. The presence of FBNs likely contributed to the successful adaptation of plants to changing environmental conditions and increased resilience [33].

Another stabilizing factor present in Au-NP-treated plants is MFP1. MFP1 is known to be involved in adaptation to varying light conditions, for instance, by facilitating the movement of LHCs during state transitions or by regulating gene expression. However, the physiological function of MFP1 in plants remains incompletely understood [34].

Importantly, the proteomic data obtained for Au-NP-treated plants, particularly with respect to DRPs involved in photosynthesis, were corroborated by physiological measurements. Experiments showed that the leaf length of treated plants was 18% longer than in the control. It can be assumed that the increase in growth under the influence of nanopriming requires high energy costs, so the sugar content in the treated plants was reduced. Specifically, nanoparticle priming led to a significant increase in chloroplast area, chlorophyll content, and photosynthetic activity—by approximately 30% (Table 1, Figure 8).

### 3.2. Generation of Precursor Metabolites and Energy

Enrichment of the biological process Generation of precursor metabolites and energy (GO:0006091) was higher in the control group. ATP synthase plays a crucial role in plant growth and development by providing cellular energy (Figure 6). In the control plants, the expression of five ATP synthase subunit alpha proteins—responsible for regulatory functions—was threefold higher compared to Au-NP-treated plants. Conversely, the expression of three ATP synthase subunit beta proteins, which contain the catalytic sites, was slightly elevated in Au-NP-treated wheat plants. Notably, the expression of two ATP synthase subunit gamma proteins was more than 15-fold higher in Au-NP-treated plants. This subunit is essential for regulating enzyme activity, particularly in inhibiting ADP binding and modulating redox responses. Moreover, the γ-subunit regulates intracellular ATP levels in response to changes in light conditions [35].

### 3.3. Transmembrane Transport

According to the proteomic analysis, three translocons at the inner envelope membrane of chloroplasts 110 (TIC110) were identified exclusively in Au-NP-treated plants and were absent in the control group (Figure 3c). These proteins are known to participate in protein import into chloroplasts [36] and are localized on the inner envelope membrane. TIC110 is essential for the translocation of nuclear-encoded proteins involved in core plastidial metabolic processes, such as amino acid and lipid biosynthesis. These complexes play a key role in plastid-type transitions, including the shift from chemoautotrophic to photoautotrophic metabolism during seedling development, and facilitate interactions with other organelles. Some inner membrane proteins are also thought to anchor plastids to cytoskeletal elements and mediate membrane contact sites with organelles such as the endoplasmic reticulum. Studies have shown that mutations in the Tic110 gene can impair chloroplast metabolism, disrupt electron transport, and in the homozygous state result in embryonic lethality, highlighting its essential role in plastid biogenesis [37]. It is plausible that the upregulation of TIC110, along with increased expression of CBPs, CP43, PsaA, PsaB, RuBisCO, ATP synthase gamma subunit, HCF136, FBNs, and MFP1, contributed to enhanced metabolic activity in the chloroplasts of Au-NP-treated plants.

Proteomic analysis revealed that the expression of two H(+)-transporting two-sector ATPases was twofold higher in Au-NP-treated plants compared to the control group, which contained three such proteins (see Appendix A). This enzyme plays a key role in ion transport, generating and maintaining a proton electrochemical gradient from the extracellular space into the cell. This gradient is subsequently utilized by secondary active transporters to import solutes into the cell, thereby facilitating nutrient uptake. Additionally, H+-ATPase is essential for plant responses to environmental stimuli, such as leaf movement.

### 3.4. Oxidative Stress

A recent study investigating the mechanisms underlying the reduction of Au or Ag during the “green synthesis” of nanoparticles in plants [16] demonstrated the activation of several proteins regulated by oxidative stress, including glutathione S-transferases, two thioredoxin-like proteins, and lactoylglutathione lyase (glyoxalase I). In our study, other proteins associated with oxidative stress protection were identified. One of them was a CBS domain-containing protein (CDCP), which plays a significant role in plant development and in responses to biotic and abiotic stresses. CDCPs are known to enhance plant tolerance to salinity, heavy metals, and oxidative stress, and are also involved in the regulation of thioredoxin activation. This regulation helps control intracellular H_2_O_2_ levels and modulates plant growth and development. Moreover, CDCPs may protect cells from genotoxic stress [38,39].

Another identified protein was a thioredoxin domain-containing protein (CDSP32), a 32-kDa chloroplast-localized protein. CDSP32 is known to protect chloroplast structures from oxidative damage during drought stress. According to previous studies, it also protected fructose-1,6-bisphosphate aldolase from oxidative damage induced by methyl viologen, tert-butyl hydroperoxide, and low temperatures under high light intensity [40]. Furthermore, thioredoxins have been shown to modulate chloroplast ATP synthase and regulate the stability of the photosynthetic apparatus in darkness, potentially by controlling protein and ion transport across the thylakoid membrane [41].

In our analysis, two heme peroxidase family domain-containing proteins were detected in Au-NP-treated plants, which are likely APX. Biochemical assays confirmed elevated APX activity in Au-NP plants (Figure 10). An increase in the activity of ascorbate peroxidase, which is part of the ascorbate-glutathione cycle in chloroplasts, also indicates an increase in the intensity of photosynthesis. The activity of other antioxidant enzymes was either unchanged (SOD) or showed non-significant increases (CAT, POX).

## 4. Materials and Methods

### 4.1. The Synthesis and Description of Au-NPs

Gold nanospheres were produced using the citrate method [42]. 250 mL of a 0.01% aqueous chloroauric acid solution was heated to 100 °C in an Erlenmeyer flask using a magnetic stirrer and a reflux water condenser. After adding 7.75 mL of 1% aqueous sodium citrate solution, the mixture was boiled for an additional half hour, or until a red sol was produced. Freshly prepared aqueous colloidal Au-NP solution was transferred into sterile glass vials with tight-fitting lids and stored at 4 °C.

### 4.2. Plant Material and Growth Conditions

Wheat plant seeds (*Triticum aestivum* L., Poaceae) were used in this study. The seeds genotype Zlata were supplied by the Federal Research Center “Nemchinovka” (Novoivanovskoye, Russia). The seeds were germinated in Au-NPs solutions (20 g/mL) for 24 h. After that, the seeds were transferred in glasses with distilled water in temperature camera (temperature of 22 °C, photoperiod of 16 h, illumination 100 μmol photons m^−2^ s^−1^). Fluorescent lamps OSRAM L 80W/640 (Osram, Smolensk, Russia) were used to illuminate the plants. After reaching 10 days seedlings were used for analysis.

### 4.3. Protein Extraction and Enzymatic Hydrolysis for Proteomic Analysis

Wheat seedlings were dissected into shoots and roots. For tissue homogenate, shoots or roots from all plants in each group were frozen in liquid nitrogen and ground in a CryoMill M-400 (Retsch, Haan, Germany; 3 cryocycles of 30 Hz for 30 s/5 Hz for 30 s).

Six samples from each group were prepared for MS analysis (an average homogenate of 2 biological repeats per group was prepared, where a biological repeat is one container containing N plants in each group). A homogenized tissue (1 g) was resuspended in 5 mL of SDS-phenol buffer (1% SDS (Bio-Rad, Hercules, CA, USA) in 50 mM Tris (Bio-Rad, Hercules, CA, USA) pH 7.5 and phenol (Vector, Russia) (pH 8.0) at a 1:1 (*v*:*v*) ratio) [43] and lysed by sonication (90% amplitude, 1-s on/1-s off cycle, 1 min and 40% amplitude, 1-s on/1-s off cycle, 1 min) (QSonica Q125, Newtown, CT, USA). The lysate was incubated at room temperature for 1 h with constant shaking, followed by centrifugation at 3400 rcf 20 min at 4 °C (Eppendorf, Hamburg, Germany). The aqueous phase was removed, the phenol phase was transferred to a clean tube and washed by gentle shaking for 5 min with an equivalent volume of water, then centrifuged at 3400 rcf 20 min at 4 °C (Eppendorf, Hamburg, Germany). The aqueous phase was removed, and a 5-fold volume of 0.1 M ammonium acetate (Honeywell, Offenbach am Main, Germany) in methanol followed by overnight incubation at −20 °C was added to the phenol phase. Samples were centrifuged at 10,700 rcf for 20 min at 4 °C, the supernatant was removed, the pellet was washed with 100% pre-chilled acetone (Khimmed, Moscow, Russia) by gentle vortexing followed by centrifugation at 10,700 rcf for 20 min at 4 °C, the supernatant was removed, the pellet was dried and redissolved in 8 M Urea (Sigma-Aldrich, St. Louis, MO, USA). The dissolved proteins were subjected to additional chloroform-methanol extraction [44]. The resulting proteins were dissolved in 1M Urea and concentrations were measured using the Pierce BCA kit (Thermo Fisher Scientific, Waltham, MA, USA). Protein disulfide bonds were reduced with 10 mM dithiothreitol (neoFroxx GmbH, Einhausen, Germany) and alkylated with 10 mM iodoacetamide (Sigma-Aldrich, St. Louis, MO, USA). The resulting protein preparations were incubated for 18 h with trypsin (Promega, Fitchburg, WI, USA; for HPLC-MS) added at a 1:50 (*w*/*w*) ratio. Enzymatic digestion was terminated by the addition of TFA, 1% *v*/*v* (Pallav, Mumbai, India) to the samples. The samples were desalted using Copure C18 SPE cartridges (Biocomma, Shenzhen, China) and dried using a vacuum concentrator (Eppendorf Concentrator Plus, Eppendorf, Hamburg, Germany) at 45 °C and stored at −40 °C until the LC-MS analysis.

### 4.4. HPLC-MS1 Analysis

LC-MS1 experiments for samples were performed using Orbitrap Q Exactive HF mass spectrometer (Thermo Fisher Scientific, Waltham, MA, USA) coupled to UltiMate 3000 LC system (Thermo Fisher Scientific, Dreieich, Germany). Luna Column (C18, 100 Å, 5 µm, 2.00 mm × 150 mm) (Phenomenex, USA) was employed for separation. Mobile phases were as follows: (A) 0.1% formic acid (FA) in water; (B) 80% ACN, 0.1% FA in water. The gradient was from 5% to 35% phase B in 19,6 min at 0.4 mL/min. Total method time including column washing and equilibration, was 36 min. The data were acquired in the MS1-only spectrum acquisition mode at 375–1500 m/z at a resolution of 120,000; the maximum automatic gain control (AGC) value was 4 × 10^5^; the maximum charge accumulation time was 50 ms. Samples were resuspended in water and quantities of 7 ug were loaded per injection.

### 4.5. Proteomic Data Processing

Raw files were converted into mzML format using ThermoRawFileParser, version 1.4.5 [45]. Peptide feature detection was performed using the Biosaur2 software, version 0.2.16 [46]. The proteomic search engine ms1searchpy [19] was used for protein identification. Searches were performed against the UniProt TrEMBL protein database (*T. aestivum* organism, taxon ID: 4565). Parameters for the search were as follows: mass tolerance for precursors was ±8 ppm, carbamidomethylation of cysteine as a fixed modification, 0 missed cleavages, and other parameters were default values. The list of identified proteins was filtered to 5% false discovery rate (FDR_TD_) using the target-decoy method [47]. The program Diffacto [48] was used for the statistical analysis of protein expression based on the intensities of peptide ions in the mass spectra. For the results obtained with the Diffacto program, differentially regulated proteins were selected according to the following criteria: FDR < 3 × 10^−38^ and |log2FC| ≥ 3.0 (shoot/roots comparison), where FDR is the *p*-value with the correction for multiple comparisons by the Benjamini–Hochberg procedure [49] and FC is the fold change in protein concentration. The results of quantification were visualized with the QRePS program, version 1.2.0 [50]; the activity metrics of biological processes were calculated based on the proteomic data as described previously [50]. Protein functions were annotated with DeepGO-SE, version 1.0.0 [51]; GOATOOLS, version 1.3.1 [52] was used for the GO analysis with a custom annotation database.

### 4.6. Ultrastructure of Leaf Cells

For electron microscopy, the samples from middle parts of 3–4 leaves were fixed by the standard method for 4 h in 2.5% glutaraldehyde in 0.1 M phosphate buffer (pH 7.4) at 0–4 °C with postfixation in 2% OsO_4_ [53]. They were dehydrated in an ethanol series (20–100%) and the material was embedded in Epon-812. Ultrathin slices of leaves were prepared using LKB-3 ultramicrotome (LKB, Luleå, Sweden). The slices were observed under LIBRA120 electron microscope (ZEISS, Oberkochen, Germany). Morphometric analysis of the ultrastructure was performed on cells of the first subepidermal layer of mesophyll by program ITEM 5.0 and Axion Vision 4.8.

### 4.7. CO_2_-Gas Exchange

The study of CO_2_-gas exchange was measured in an open-type unit equipped with a URAS 2T IR gas analyzer (Hartmann und Braun, Frankfurt am Main, Germany). The measurements of gas exchange included the determination of the net CO_2_ assimilation rates and dark respiration, both were expressed in mg CO_2_/g dry weight per hour.

### 4.8. Photosynthetic Pigments Content

The content of chlorophyll a (Chl a), chlorophyll b (Chl b), and total carotenoids (Car) in the leaves was determined spectrophotometrically at wavelengths corresponding to the absorption maxima of Chl a, Chl b and Car in an 80% acetone (Reachem, Moscow, Russia) solution—663 nm, 646 nm and 470 nm, respectively. Pigment concentrations were calculated using the following formulas [54]:Ca = 12.21D663 − 2.81D646Cb = 20.13D646 − 5.03D663Ccar = (1000D470 − 3.27Ca − 104Cb)/198

The pigment content was expressed as mg/g dry weight of leaves.

### 4.9. Soluble Sugars (Glucose, Fructose, and Sucrose) Content

Weighed samples (500 mg) of leaves were fixed with 96% boiling ethanol (Reachem, Moscow, Russia). Tissue was homogenized, and sugars were extracted with 80% ethanol three times. In the obtained extracts the content of fructose was determined according to Roe, followed by conversion to sucrose content [55].

### 4.10. Malondialdehyde (MDA) Content

Lipid peroxidation (LPO) level in the leaves was determined as the content of MDA as described by Heath and Packer [56] with minor modifications. Leaf sample (≈300 mg) was homogenized in 5 mL of 0.35 M NaCl (Reachem, Moscow, Russia) in 0.1 M Tris-HCl buffer (Reachem, Moscow, Russia), pH 7.6. The homogenate (3 mL) was mixed with 2 mL of 0.5% thiobarbituric acid (Merck SA, Darmstadt, Germany) in 20% trichloroacetic acid (Reachem, Moscow, Russia) and then heated at 100 °C for 30 min, cooled, and filtered. The extraction medium containing the reagent was used as a control. A Genesys 10 UV spectrophotometer (Termo Electron Corporation, Waltham, MA, USA) was used in the study. Measurements were corrected for nonspecific absorbance by subtracting the values obtained at 532 nm and 600 nm. The content of MDA was calculated in μM/g dry weight of leaves.

### 4.11. Extraction Soluble Proteins and Activity of Antioxidant Enzymes

Leaf samples (500 mg) were homogenized in isolation medium (50 mM Tris-HCl, pH 7.6, 3 mM EDTA, 250 mM sucrose, 3.6 mM cysteine, 5 mM ascorbic acid, 3 mM MgCl2, 2 mM DTT, 2 mM PMSF) (Reachem, Moscow, Russia), and centrifuged for 20 min at 16,000× *g*. The extract was purified on a PD-10 midiTrap G-25 column (GE Healthcare, Chicago, IL, USA). The protein content in the supernatant was determined spectrophotometrically at 562 nm using the Bicinchoninic Acid Kit for Protein Determination (BCA1-1KT, Sigma-Aldrich, Saint Louis, MO, USA), according to the manufacturer’s protocol.

Superoxide dismutase (SOD, EC 1.15.1.1) activity was determined using the method [57] based on the generation of superoxide radicals in the course of riboflavin photooxidation that is enhanced by an indicator trap (nitroblue tetrazolium). The reaction mixture contained 1.5% L-methionine, 0.14% NBT and 1% Triton X100 in a ratio of 3:1:0.75. Optical density measurements were performed at 560 nm. The inhibition of formazan production by 50% was recognized as a unit of SOD activity, which was converted to 1 g of protein.

Catalase activity (CAT, EC 1.11.1.6) was measured by the rate of the H_2_O_2_ decomposition reaction [58]. The final reaction comprised of 2.8 mL Tris-HCl buffer, pH 7.6, 0.3 mL of enzyme extract and 0.3 mL of H_2_O_2_ (3 mM) for 3 min. Observing the depletion of H_2_O_2_ in accordance with a drop in absorbance at 240 nm was used to measure CAT activity in the supernatant. CAT activity was expressed as μM H_2_O_2_/g protein × min.

Guaiacol peroxidase activity (POX, EC 1.11.1.7) was determined according to Kumar and Knowles [58] with modifications. The method is based on oxidation of guaiacol to tetraguaiacol. The reaction mixture contained 0.3 mL of 0.15 mM guaiacol, 0.9 mL of Tris-HCl buffer (10 mM, pH 7.6) and 0.3 mL of the supernatant. Before optical density measurement, 0.3 mL of 0.1 mM H_2_O_2_ was added to the cuvette. The optical density was measured at 470 nm. The reaction proceeded at 25 °C for 1 min at intervals of 10 s. POX activity was expressed as μM guaiacol/g protein × min.

The activity of ascorbate peroxidase (APX, EC 1.11.1.11) was determined according to Nakano and Asada [59] with modifications. The method is based on the measurement of the rate of ascorbate decrease in the reaction catalyzed by ascorbate peroxidase. The reaction mixture included 0.3 mL of protein extract, 100 μL of 5 mM ascorbate, 50 μL of 0.1 mM EDTA, and 2.75 mL of Tris-HCl buffer (10 mM, pH 7.6). Before optical density measurement, 100 μL of 0.1 mM H_2_O_2_ was added to the cuvette. Optical density was measured at 290 nm for 1 min at intervals of 10 s. APX activity was expressed as μM ascorbate/g protein × min.

### 4.12. RNA Isolation and RT-PCR (qRT-PCR)

Total RNA was isolated from a sample of leaves (50 mg) using the Spectrum Plant Total RNA Kit (Sigma-Aldrich, St. Louis, MO, USA) with a modification of the homogenization stage. The quality and quantity of purified RNA were determined using a NanoDrop 2000 spectrophotometer (Thermo Fisher Scientific, Waltham, MA, USA) and analyzed electrophoretically in a 2% agarose gel. To remove residual genomic DNA impurities, total RNA preparations were treated with DNase I (Thermo Fisher Scientific, Waltham, MA, USA). For cDNA synthesis, we used RevertAid reverse transcription kits (Thermo Fisher Scientific, Waltham, MA, USA). RT-qPCR was performed on a CFX96 Touch amplifier (Bio-Rad, USA) using the intercalating dye SYBR Green I (ZAO Evrogen, Moscow, Russia). The reaction mixture for quantitative PCR in a volume of 25 μL contained 5 μL of qPCRmix HS SYBR (JSC Evrogen, Moscow, Russia), 0.2 μmol of each primer, and 15 ng of cDNA template. The following amplification conditions were used: 95 °C for 5 min, then 40 cycles: 95 °C for 15 s, 60 °C for 30 s, and 72 °C for 30 s.

Nucleotide sequences of primers for amplification of target genes of small (rbcS) and big (rbcL) ribulose-1,5-bisphosphate carboxylase/oxygenase (RuBisCo) subunits were borrowed from publication [60]. Gene-specific primers for amplification of reference gene (TaRP15) were selected (https://www.ncbi.nlm.nih.gov/tools/primer-blast/, accessed on 13 March 2024) and OligoAnalyzer™ Tool (https://eu.idtdna.com/pages/tools/oligoanalyzer, accessed on 13 March 2024) online resources. Primers were designed to bind to all three wheat sub-genomes. The transcript levels were normalized to the expression of the reference gene TaRP15 (RNA polymerase I, II, and III, 15 kDa subunit). The primers used are presented in Table 4. The amplification results were processed using the Pfaffl method [61]. Each qRT-PCR reaction was performed in three biological and two technical replicates.

### 4.13. Gold Content

Determining gold content in seeds, roots, and leaves of 10-day-old wheat seedlings was carried out using an Agilent inductively coupled plasma atomic emission spectrometer (Agilent Technologies, Santa Clara, CA, USA) and a Bruker 820-MS ICP-MS inductively coupled plasma mass spectrometer (Bruker, Bremen, Germany).

### 4.14. Statistical Analysis

All experiments were carried out in three to six biological and three to four analytical replicates. The tables and figures present arithmetic means and their standard errors. The statistical significance of differences between means was assessed using ANOVA (Tukey’s test was used) built into the Microcal Origin 7.0 graphical mathematical package. The article discusses only values that are statistically significant at *p* < 0.05.

## 5. Conclusions

This study convincingly demonstrated that nanopriming with Au-NPs at the selected concentration did not induce any toxic effects or stress responses in wheat plants. This conclusion is strongly supported by our proteomic analysis, which revealed that the majority of DRPs identified in Au-NP-treated wheat plants were associated with constitutive metabolic processes. Even under optimal growth conditions, Au-NPs treatment resulted in a global reorganization of the PSA, both at the cellular ultrastructural level and at the level of biochemical processes. This reorganization appeared to regulate carbon flux during photosynthesis, likely through enhanced CO_2_ assimilation, more efficient light energy utilization, and regulation of the PSA stability in the dark, potentially via control of protein and ion transport across the thylakoid membrane. These changes were accompanied by an increased abundance of proteins involved in oxidative stress response and energy production. While control plants appeared to activate certain mechanisms to increase photosynthetic efficiency—such as enhanced RuBisCO activity through upregulation of carbonic anhydrases, RuBisCO activases, and regeneration of RuBP via elevated sedoheptulose-1,7-bisphosphatase activity—Au-NP-treated plants exhibited intrinsically enhanced photosynthetic performance as a direct result of nanoparticle exposure. In the long term, such physiological and molecular changes may contribute to increased crop productivity and possibly increased resistance to both abiotic and biotic stresses. This supports the potential for broad application of seed nanopriming technology with gold nanoparticles in industrial agriculture.

## Figures and Tables

**Figure 1 ijms-26-07608-f001:**
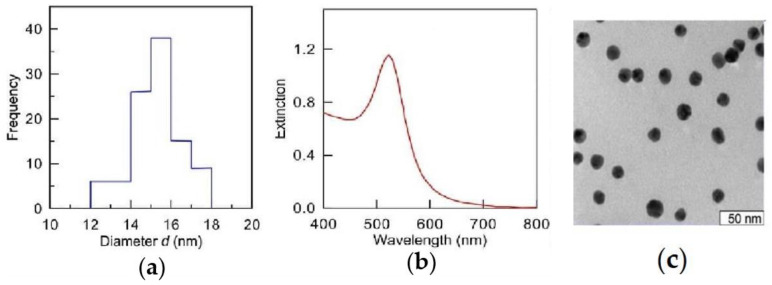
Size distribution (**a**), absorption spectrum (**b**), and TEM image (**c**) of gold nanoparticles with an average diameter of 15 nm.

**Figure 2 ijms-26-07608-f002:**
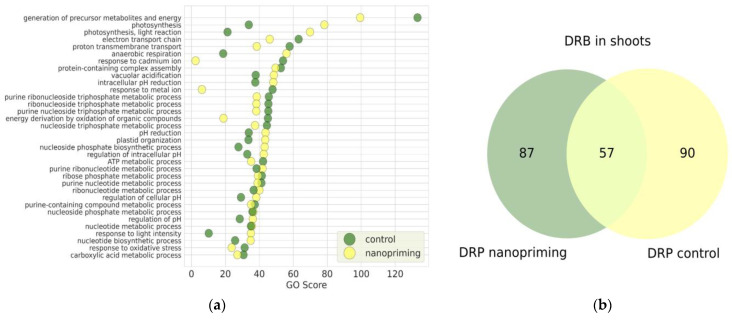
(**a**) Enrichment of biological processes (34) in wheat based on DRP identified by DirectMS1Quant method for different seed treatments: green—seedlings grown from the seeds treated with distilled water (control), yellow—seedlings grown from the seeds treated with Au-NPs. GO score = log_10_ (E) × |log_10_ (FDR)|, where E is the enrichment of biological process, FDR is the statistical significance of enrichment with correction for multiple comparisons by the Benjamini-Hochberg procedure; (**b**) Venn diagram illustrating the overlap of differentially regulated proteins in treated Au-NPs and control. The numbers represent the unique and shared differentially regulated proteins.

**Figure 3 ijms-26-07608-f003:**
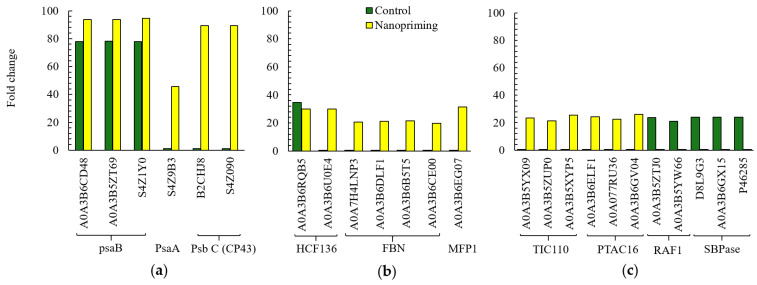
Some of the DRP participated in photosynthesis (GO:0015979), photosynthesis, light reaction (GO: 0019684), and plastid organization (GO:0009657). (**a**) photosystem I (A0A3B6CD48, A0A3B5ZT69, S4Z1Y0, S4Z9B3), PS II chlorophyll proteins of 43 kDa (CP43: B2CHJ8, S4Z090); (**b**) photosystem II stability/assembly factor HCF136 (A0A3B6RQB5), plastid lipid-associated protein/fibrillin conserved (FBN, A0A7H4LNP3, A0A3B6DLF1, A0A3B6B5T5, A0A3B6CE00), MAR-binding filament-like protein 1 (MFP1, A0A3B6EG07); (**c**) translocon at the inner envelope membrane of chloroplasts 110 (TIC110, A0A3B5YX09, A0A3B5ZUP0, A0A3B5XYP5), plastid transcriptionally active 16 (PTAC16, A0A3B6ELF1, A0A077RU36, A0A3B6GV04), Rubisco accumulation factor 1 C-terminal domain-containing protein (RAF1: A0A3B5ZTJ0, A0A3B5YW66), Sedoheptulose-1,7-bisphosphatase (SBPase, D8L9G3, A0A3B6GX15, P46285).

**Figure 4 ijms-26-07608-f004:**
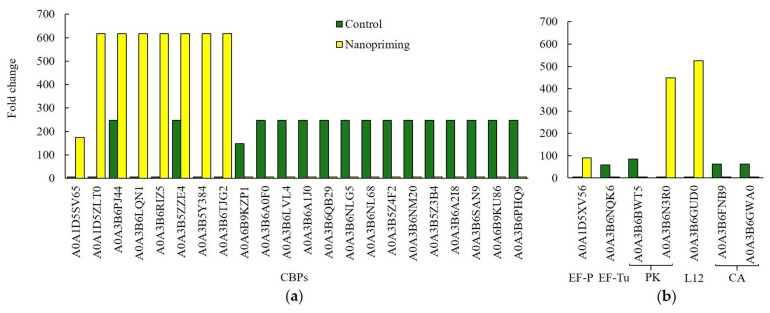
Some of the DRP participated in photosynthesis, light reaction (GO: 0019684), generation of precursor metabolites and energy (GO:0006091), photosynthesis (GO:0015979), primary metabolic process (GO:0044238). (**a**) chlorophyll a-b binding proteins (CBPs); (**b**) elongation factor (A0A1D5XV56, A0A3B6NQK6), non-specific serine/threonine protein kinase (PK: A0A3B6BWT5, A0A3B6N3R0), 50S ribosomal protein L12 (A0A3B6GUD0), carbonic anhydrase (CA: A0A3B6FNB9, A0A3B6GWA0).

**Figure 5 ijms-26-07608-f005:**
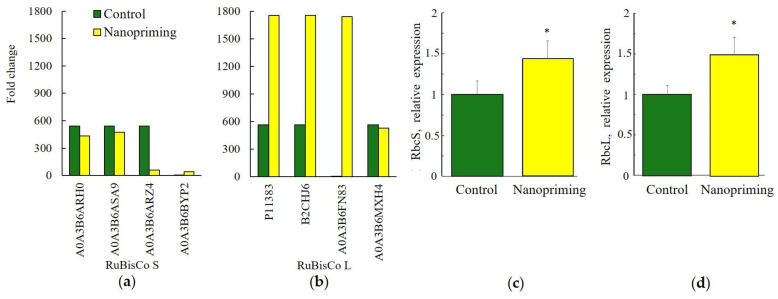
Some of the DRP participated in photosynthesis, light reaction (GO: 0019684). (**a**) ribulose bisphosphate carboxylase S (RuBisCo S, A0A3B6ARH0, A0A3B6ASA9, A0A3B6ARZ4, A0A3B6BYP2); (**b**) ribulose bisphosphate carboxylase L (RuBisCo L, P11383, B2CHJ6, A0A3B6FN83, A0A3B6MXH4); (**c**) effect of nanopriming on Ribulose 1,5-bisphosphate carboxylase/oxygenase (RuBisCO) small and large; (**d**) subunits relative gene expression in leaves of wheat. The values that significantly differ at *p* < 0.05 (Tukey’s *t*-test) are denoted by an asterisk.

**Figure 6 ijms-26-07608-f006:**
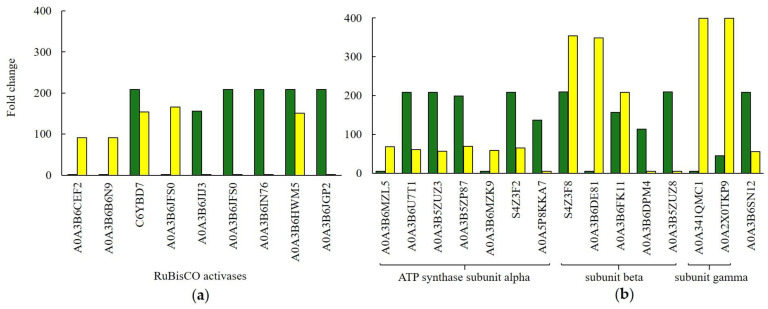
Some of the DRP participated in proton motive force-driven ATP synthesis (GO:0015986), hexose biosynthetic process (GO:0019319). (**a**) ribulose bisphosphate carboxylase/oxygenase activase (A0A3B6CEF2, A0A3B6B6N9, C6YBD7, A0A3B6JFS0, A0A3B6JIJ3, A0A3B6JFS0, A0A3B6IN76, A0A3B6HWM5, A0A3B6JGP2); (**b**) ATP synthase (A0A3B6MZL5, A0A3B6U7T1, A0A3B5ZUZ3, A0A3B5ZP87, A0A3B6MZK9, S4Z3F2, A0A5P8KKA7, S4Z3F8, A0A3B6DE81, A0A3B6FK11, A0A3B6DPM4, A0A3B5ZUZ8, A0A341QMC1, A0A2X0TKP9, A0A3B6SN12).

**Figure 7 ijms-26-07608-f007:**
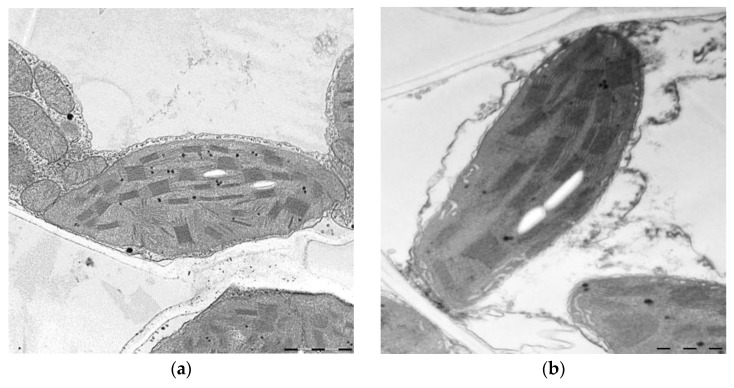
Effect of nanopriming on the ultrastructure of chloroplasts in wheat leaves: (**a**) control plants; (**b**) nanopriming. Bar 1 µm.

**Figure 8 ijms-26-07608-f008:**
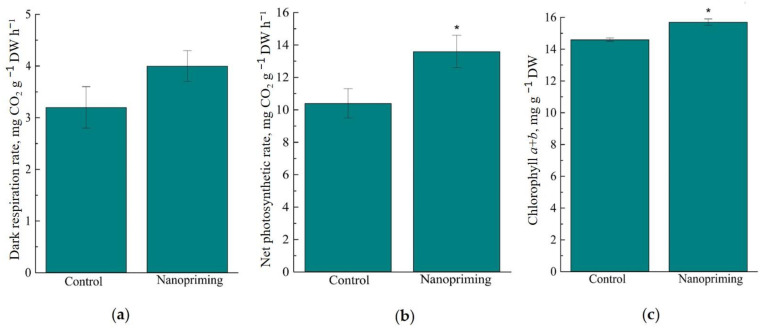
Effect of nanopriming Au-NPs on some biochemical and physiological processes in wheat. (**a**) respiratory rate; (**b**) intensity of photosynthesis; (**c**) chlorophyll content. The values that significantly differ at *p* < 0.05 (Tukey’s *t*-test) are denoted by an asterisk.

**Figure 9 ijms-26-07608-f009:**
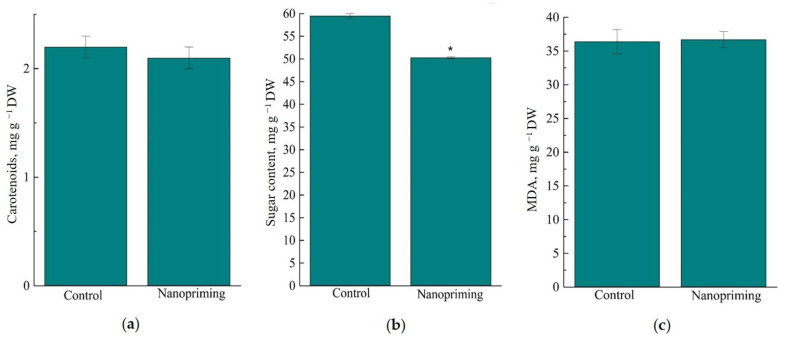
Effect of nanopriming Au-NPs on some biochemical and physiological processes in wheat. (**a**) Carotenoid content; (**b**) soluble sugar content; (**c**) lipid peroxidation intensity. The values that significantly differ at *p* < 0.05 (Tukey’s *t*-test) are denoted by an asterisk.

**Figure 10 ijms-26-07608-f010:**
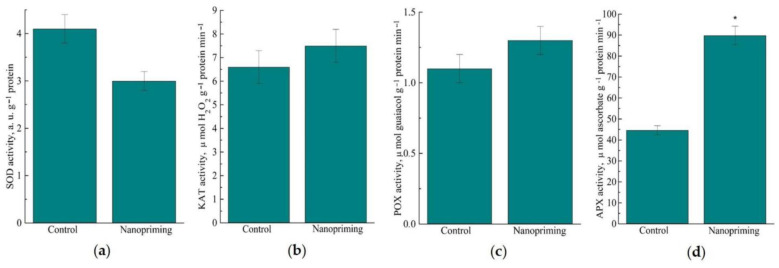
Effect of nanopriming on the activity of antioxidant ferments in leaves of wheat. (**a**) Superoxide dismutase (SOD) activity; (**b**) catalase (CAT) activity; (**c**) guaiacol-dependent peroxidase III class (POX) activity; (**d**) ascorbate peroxidase (APX) activity. The values that significantly differ at *p* < 0.05 (Tukey’s *t*-test) are denoted by an asterisk.

**Table 1 ijms-26-07608-t001:** Effect of nanopriming Au-NPs on the growth of wheat, area of the main organelles, and the ultrastructure of chloroplasts in wheat leaves.

Indicator	Control	Nanopriming
Seed germination, %	77 ± 2	76 ± 2
Leaf length, mm	136 ± 3	154 ± 3 *
Area of peroxisome, µm^2^	0.25 ± 0.02	0.18 ± 0.02 *
Area of mitochondria, µm^2^	0.28 ± 0.01	0.16 ± 0.02 *
Number of cristae per 1 µm^2^, pcs	48 ± 2	49 ± 3
Area of chloroplast, µm^2^	5.10 ± 0.1	5.70 ± 0.1 *
Number of grana per 10 µm^2^ of chloroplast, pcs	51 ± 1	56 ± 1 *
Number of plastoglobules per 10 µm^2^ of chloroplast, pcs	37 ± 4	36 ± 1
Number of thylakoids in grana, pcs	7 ± 0.1	7 ± 0.2
Area of starch grain, µm^2^	0.04 ± 0.006	0.07 ± 0.02 *
Area of starch in chloroplast, %	0.70 ± 0.1	1.20 ± 0.1 *

* The values that significantly differ at *p* < 0.05 (Tukey’s *t*-test) are denoted by an asterisk.

**Table 2 ijms-26-07608-t002:** Effect of nanopriming on the distribution of grana by the number of thylakoids in chloroplasts of wheat leaves.

Number of Thylakoids in Grana	Number of Grana in Chloroplast, %	
	Control	Nanopriming
2–6	48 ± 1	56 ± 2 *
7–16	49 ± 1	40 ± 2 *
17–21	3 ± 0.6	4 ± 1

* The values that significantly differ at *p* < 0.05 (Tukey’s *t*-test) are denoted by an asterisk.

**Table 3 ijms-26-07608-t003:** Content of gold in organs of wheat, mg/kg of dry weight.

Organs	Control	Nanopriming
Seeds	<0.05	3.8 ± 0.1
Roots	<0.05	1.6 ± 0.1
Leaves	<0.05	0.3 ± 0.1

**Table 4 ijms-26-07608-t004:** Primers for PCR analysis.

Gene	Primer	Primer Sequences
*TaRP15*	Forward Reverse	TCATTGTGGAGGACTCGTGG GCAGACATAGCCCACACAT
*rbcS*	Forward Reverse	GGATTCGACAACATGCGCCAGG ATATGGCCTGTCGTGAGTGAGC
*rbcL*	Forward Reverse	ACCATTTATGCGCTGGAGAGACC CAAGTAATGCCCCTTGATTTCACC

## Data Availability

The mass spectrometry-based proteomics data have been DRPosited to the ProteomeXchange Consortium via the PRIDE (PXD065948) partner repository with the dataset identifier.

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
