# Peer review of "Functional Express Proteomics for Search and Identification of Differentially Regulated Proteins Involved in the Reaction of Wheat (Triticum aestivum L.) to Nanopriming by Gold Nanoparticles"

_ijms, 2025, doi:10.3390/ijms26157608_

Round 1

Reviewer 1 Report

Comments and Suggestions for Authors

The authors pre-treated wheat seeds with nano-gold particles and discovered that this treatment could enhance the photosynthesis of wheat during the seedling stage. The underlying mechanism involves alterations in the ultrastructure of chloroplasts and an improvement in the antioxidant system. This manuscript is really interesting and well-written. I think it’ll be good to go once a few revisions are made.

Comments

  1. The characterization of nano-gold particles should be presented as Figure 1 within the main body of the article, rather than merely being described briefly in the Materials and Methods section.
  2. Do nano-gold particles exert any influence on the germination of wheat seeds and the growth rate of seedlings?
  3. The introduction provides a good foundation, but it could be expanded to offer a more comprehensive overview. Adding recent research advancements in nano-agriculture would greatly enhance its depth and relevance.
  4. Which gene was utilized as an internal control to normalize the expression levels of the target genes? Please provide in Table 4.

Author Response

Dear Reviewer,

We thank you for your attention to our work and its high appreciation. Your questions and comments help us to better understand our material and improve the manuscript. We have made the changes proposed by the reviewer (marked in red in the manuscript). Answers to the comments of the Reviewer are summarized below.

  1. The characterization of nano-gold particles should be presented as Figure 1 within the main body of the article, rather than merely being described briefly in the Materials and Methods section. 

    Response. Changes were made in the manuscript - the characterization of Au-NPs is presented as Figure 1 within the Results section (lines 116-129).

  2. Do nano-gold particles exert any influence on the germination of wheat seeds and the growth rate of seedlings?

    Response. We thank the reviewer for his interest. We  included in the manuscript the data about the growth indicators of wheat (Table 1). Au-NPs did not affect the germination of seeds, but noticeably enhanced the growth of wheat (according to the length of 1 leaf).

  3. The introduction provides a good foundation, but it could be expanded to offer a more comprehensive overview. Adding recent research advancements in nano-agriculture would greatly enhance its depth and relevance.

    Response. Changes were made in the Introduction (lines 43 – 83).

  4. Which gene was utilized as an internal control to normalize the expression levels of the target genes? Please provide in Table 4. 

    Response. Made additions to the real-time methodology (lines 660 – 667).

    We express our gratitude to the esteemed Reviewer once more for his consideration of our manuscript,

    Best wishes,

    Natalia Naraikina and coauthors.

Reviewer 2 Report

Comments and Suggestions for Authors

The manuscript presents an integrative proteomic investigation into the physiological and molecular changes in wheat seedlings primed with gold nanoparticles (Au-NPs). The topic is timely and relevant, and the overall approach is well motivated. However, several issues regarding data interpretation, methodological clarity, and figure presentation need to be addressed before the manuscript can be considered for publication.

First, there are some inconsistencies and gaps in logic that should be clarified. For instance, the expression of RuBisCO large subunit is significantly elevated in Au-NP-treated plants, whereas RuBisCO activase expression is higher in the control group. While the authors briefly mention the presence of GLO5 as a possible inhibitor of activase, the biological implication of this contradictory pattern is not sufficiently developed. Additional explanation or references would help resolve this apparent inconsistency.

Furthermore, the manuscript reports an increase in photosynthetic rate alongside a 16% decrease in soluble sugars in Au-NP-treated plants. This finding is counterintuitive and merits further discussion. Are sugars being mobilized more rapidly? Is there enhanced translocation to roots or other sinks? Currently, this result is stated without context, leaving readers with unresolved questions.

The authors also suggest that Au-NP treatment confers enhanced stress tolerance, but this conclusion is mainly based on an observed increase in APX activity, while no significant changes are found in SOD, CAT, or lipid peroxidation (MDA content). Given the limited and mixed evidence, this claim should be either toned down or supported with additional data.

From a methodological standpoint, the integration between proteomic and transcriptomic layers is insufficient. Although qRT-PCR data are presented for RuBisCO genes, other differentially expressed proteins such as PsaA, PsaB, CP43, and TIC110 were not validated at the transcript level. This weakens the argument that the proteomic results are well supported at the transcriptional level.

Moreover, while the study focuses heavily on shoot-level analyses, root responses—which are likely the primary site of early exposure following seed priming—are underexplored. Including more physiological or proteomic data from root tissue would make the conclusions more balanced.

Regarding experimental design and reporting, there is ambiguity around the number of biological versus technical replicates used for the LC-MS proteomics. Although the authors mention six samples and "two biological repeats," it is unclear whether the figures (Figures 2–5) represent individual replicates, averaged values, or representative samples. This should be clarified in both the methods and figure legends.

Figures 2–5, in particular, appear to be based on bar plots or schematic visualizations of protein expression, but they lack error bars, significance markers, or clear explanation of statistical analysis. It is also not stated whether these values were normalized or how they were derived. The absence of this information significantly reduces the interpretability and credibility of the visualized data. I recommend that the authors revise these figures to include error bars (if averaged), indicate significance levels (e.g., p < 0.05), and provide clearer legends describing the data source and replication.

In addition, there is noticeable redundancy across several figures. For example, PSII-related proteins are shown in both Figure 2 and Figure 3, and RuBisCO components appear in both Figure 4 and Figure 10. Streamlining these to avoid overlap would enhance clarity and reduce confusion.

Finally, while the physiological measurements are informative, the study would be strengthened by including data on whole-plant traits such as shoot/root biomass, leaf area, or growth rate. These are especially relevant when discussing productivity potential or broader agricultural implications.

In conclusion, the study addresses a relevant topic and contains valuable data, but important revisions are required to address issues in logic, data presentation, methodological transparency, and completeness of physiological interpretation.

Minor comments:

  1. Figure legends should specify whether values represent means of biological replicates, and indicate error bars and statistical significance if applicable.
  2. Bar graphs across Figures 2–5 use inconsistent scales; standardizing y-axis ranges would improve clarity.
  3. Protein identifiers such as UniProt codes are not intuitive; including gene or protein names would aid interpretation.
  4. Abbreviations like CP43, FBN, and TIC110 should be defined in full upon first use in the main text.
  5. The resolution of the TEM images in Figure 6 could be improved, and scale bars should be clearly labeled within each image panel.
  6. Units and decimal precision in Tables 1 and 2 should be harmonized for consistency and readability.
  7. The supplementary proteomics table (Table S1) is not clearly cited in the Results section; consider adding an in-text reference.
  8. Several references are missing DOIs and should be updated for completeness.
  9. The species name Triticum aestivum L. should be consistently italicized throughout the manuscript.
Comments on the Quality of English Language

The manuscript is generally readable and conveys the key scientific findings adequately. However, the English writing would benefit from thorough revision to improve clarity, conciseness, and overall flow.  Some terminology is introduced abruptly without prior definition, which may hinder comprehension for broader readers. Figure legends and certain paragraphs in the Introduction and Discussion would also benefit from clearer and more precise phrasing. A careful round of professional language editing is recommended to ensure the manuscript meets the standards of an international journal.

Author Response

Dear Reviewer,

We thank you for your attention to our work and its high appreciation. Your questions and comments help us to better understand our material and improve the manuscript. We have made the changes proposed by the reviewer (marked in red in the manuscript). Answers to the comments of the Reviewer are summarized below.

First, there are some inconsistencies and gaps in logic that should be clarified. For instance, the expression of RuBisCO large subunit is significantly elevated in Au-NP-treated plants, whereas RuBisCO activase expression is higher in the control group. While the authors briefly mention the presence of GLO5 as a possible inhibitor of activase, the biological implication of this contradictory pattern is not sufficiently developed. Additional explanation or references would help resolve this apparent inconsistency.

Response. Added information on GLO to the discussion. We suggest that this logical inconsistency is due to the fact that the increase in the expression of the large subunit of RuBisCO in nanoparticle-treated plants is associated not with rubisco activase, but with other processes. In conclusion we write: Au-NP-treated plants exhibited intrinsically enhanced photosynthetic performance as a direct result of nanoparticle exposure. Plants typically use only a small portion of sunlight in the process of photosynthesis. The theoretical maximum efficiency of solar energy conversion is approximately 11%. Treated plants can probably absorb more light (we see this from changes in photosystems I and II), which leads to an increase in the amount of light reaction products - ATP and NADPH. This, in turn, leads to an increase in the activity of RuBisCO.

Furthermore, the manuscript reports an increase in photosynthetic rate alongside a 16% decrease in soluble sugars in Au-NP-treated plants. This finding is counterintuitive and merits further discussion. Are sugars being mobilized more rapidly? Is there enhanced translocation to roots or other sinks? Currently, this result is stated without context, leaving readers with unresolved questions.

Response. In the manuscript we included data on the effect of Au-NPs on wheat growth (table 1). Experiments showed that the leaf length of treated plants was 18% longer than in the control. It can be assumed that the increase of growth under the influence of nanopriming requires high energy costs, so the sugar content in the treated plants was reduced.

The authors also suggest that Au-NP treatment confers enhanced stress tolerance, but this conclusion is mainly based on an observed increase in APX activity, while no significant changes are found in SOD, CAT, or lipid peroxidation (MDA content). Given the limited and mixed evidence, this claim should be either toned down or supported with additional data.

Response. We thank the reviewer for this correct remark. We have softened the wording in the Conclusions section by using the term "possibly" In addition, we added a phrase to the discussion: An increase in the activity of ascorbate peroxidase, which is part of the ascorbate-glutathione cycle in chloroplasts, may also indicate an increase in the intensity of oxidative processes in the PSA. Au-NP-treated plants exhibited intrinsically enhanced photosynthetic performance as a direct result of nanoparticle exposure.

Moreover, while the study focuses heavily on shoot-level analyses, root responses—which are likely the primary site of early exposure following seed priming—are underexplored. Including more physiological or proteomic data from root tissue would make the conclusions more balanced.

Response. We have not evaluated the physiological characteristics in the roots, but plan to do so in the following studies. In wheat, seeds and young shoots are used for food, so we first concentrated on studying the shoots of wheat. In addition, roots were a methodological necessity for proteome analysis: root proteins were compared with shoots to select DRPs.

Finally, while the physiological measurements are informative, the study would be strengthened by including data on whole-plant traits such as shoot/root biomass, leaf area, or growth rate. These are especially relevant when discussing productivity potential or broader agricultural implications.

Response. We thank the Reviewer for his comment. We completely agree with him. In the manuscript are inserted the data we have on the effect of Au-NPs on wheat growth (Table 1).

Regarding experimental design and reporting, there is ambiguity around the number of biological versus technical replicates used for the LC-MS proteomics. Although the authors mention six samples and "two biological repeats," it is unclear whether the figures (Figures 2–5) represent individual replicates, averaged values, or representative samples. This should be clarified in both the methods and figure legends.

Figures 2–5, in particular, appear to be based on bar plots or schematic visualizations of protein expression, but they lack error bars, significance markers, or clear explanation of statistical analysis. It is also not stated whether these values were normalized or how they were derived. The absence of this information significantly reduces the interpretability and credibility of the visualized data. I recommend that the authors revise these figures to include error bars (if averaged), indicate significance levels (e.g., p < 0.05), and provide clearer legends describing the data source and replication.

Response. Thank you for the insightful remark. Indeed, the experiment used two biological replicates and six technical replicates for each biological replicate. Quantitative analysis of protein expression was performed using the automated Difracto tool, which implements statistical data processing based on differential factor analysis of peptide abundance in each replicate. Briefly, the algorithm estimates the relative abundance of proteins by aggregating information from the peptide level and taking into account variability between replicates. The output is the fold change (FC) values for each protein, as well as the corresponding statistical significance values with the Benjamini-Hochberg correction, denoted as FDR (false discovery rate), which serves as a measure of the probability of false positive results. For subsequent interpretation of the results, threshold values for FC b FDR were set, based on which statistically significant differentially regulated proteins were selected for the compared groups (control and treatment). Thus, replicates play a key role in the statistical analysis step, ensuring the statistical reliability of FC and FDR calculations.

You are right that the bar graphs can potentially be interpreted ambiguously. However, for all proteins plotted in the Supplementary Materials, a detailed table containing FDR and FC values is provided, which ensures transparency and reproducibility of the results.

In addition, there is noticeable redundancy across several figures. For example, PSII-related proteins are shown in both Figure 2 and Figure 3, and RuBisCO components appear in both Figure 4 and Figure 10. Streamlining these to avoid overlap would enhance clarity and reduce confusion.

Response. Thank you for the insightful remark. In Figure 3a, the proteins S4Z9B3 and S4Z1Y0 were swapped so that they would follow the order of their description in the text. Double signatures (Uniprot identifier and protein name) were made in those figures where there are many proteins. The text description of the proteins shown in Figure 3c has also been moved to the corresponding figure. The components of RuBisCO are presented in one place - in Figure 5. The text description of the proteins shown in Figure 4b has been moved to the corresponding figure. Removed some proteins (lines 231-237) to make the description less redundant

Minor comments:

Bar graphs across Figures 2–5 use inconsistent scales; standardizing y-axis ranges would improve clarity.

Response: We standardized the ranges along the Y axis in Figures 3-6.

Protein identifiers such as UniProt codes are not intuitive; including gene or protein names would aid interpretation.

Response: We made double captions (uniprot identifier and protein name) on those figures where there are many different proteins.

Abbreviations like CP43, FBN, and TIC110 should be defined in full upon first use in the main text.

Response: We did it.

Units and decimal precision in Tables 1 and 2 should be harmonized for consistency and readability.

Response: We did it.

Units and decimal precision in Tables 1 and 2 should be harmonized for consistency and readability.

Response: Link added.

Several references are missing DOIs and should be updated for completeness.

Response: Links added.

The species name Triticum aestivum L. should be consistently italicized throughout the manuscript.

Response: We did it

Best wishes,

Natalia Naraikina and coauthors.

Round 2

Reviewer 1 Report

Comments and Suggestions for Authors

All of my concerns have been thoughtfully resolved, and I am pleased to recommend this for publication.